# Detection of Polymorphisms in the *MTNR1A* Gene and Their Association with Reproductive Performance in Awassi Ewes

**DOI:** 10.3390/ani11020583

**Published:** 2021-02-23

**Authors:** Giovanni Cosso, Michella Nehme, Sebastiano Luridiana, Luisa Pulinas, Giulio Curone, Chadi Hosri, Vincenzo Carcangiu, Maria Consuelo Mura

**Affiliations:** 1Department of Veterinary Medicine of Sassari, University of Sassari, Via Vienna 2, 07100 Sassari, Italy; gicosso@uniss.it (G.C.); sluridiana@uniss.it (S.L.); luisapulinas@hotmail.it (L.P.); endvet@uniss.it (V.C.); 2Department of Agriculture and Food Engineering, Faculty of Engineering, Holy Spirit University of Kaslik, Kaslik, Jounieh 446, Lebanon; michellanehme@hotmail.com; 3Department of Veterinary Medicine of Milan, University of Milan, Via dell’Università 6, 26900 Lodi, Italy; giulio.curone@unimi.it; 4Department of Veterinary Sciences, Faculty of Agriculture, Lebanese University, Dekwaneh, Beirut 14/6573, Lebanon; chadihosri@hotmail.com

**Keywords:** Awassi ewes, reproductive resumption, *MTNR1A* gene, SNPs

## Abstract

**Simple Summary:**

The purpose of the study was to explore the influence of *MTNR1A* gene polymorphisms on the reproductive performance in Awassi sheep, which is an important and widespread breed in developing Mediterranean countries. A total of 31 SNPs was detected, 5 of which caused amino acid changes. Two of the found SNPs were found to be totally linked and associated with an advanced reproductive recovery in ewes carrying the C allele. The obtained results could be useful for improving reproductive management in developing Mediterranean areas.

**Abstract:**

The economy in Mediterranean areas is tightly linked to the evolution of the sheep-farming system; therefore, improvement in ewe’s reproductive performance is essential in the developing countries of this area. *MTNR1A* is the gene coding for Melatonin receptor 1 (MT1), and it is considered to be involved in the reproductive activity in sheep. The aims of this study were: (1) identifying the polymorphisms from the entire *MTNR1A* coding region and promoter in Lebanese Awassi sheep flocks, and (2) investigating the association between the found polymorphisms and the reproductive performance, assessed as lambing rate, litter size, and days to lambing (DTL). The study was conducted in two districts of Lebanon, where 165 lactating ewes, aged 5.2 ± 1.5 years, with body condition score (BCS) 3.3 ± 0.4, were chosen and exposed to adult and fertile rams. From 150 to 220 days after ram introduction, lambing dates and litter sizes were registered. This study provided the entire coding region of the *MTNR1A* receptor gene in the Awassi sheep breed. Thirty-one single nucleotide polymorphisms (SNPs) were detected, five of which were missense mutations. The H2, H3, and H4 haplotypes were associated with lower DTL (*p* < 0.05), as well as the SNPs rs430181568 and rs40738822721, named from now on SNP20 and SNP21, respectively. These SNPs were totally linked and can be considered as a single marker. The ewes carrying the C allele at both these polymorphic sites advanced their reproductive recovery (*p* < 0.05). These results are essential for improving reproductive management and obtaining advanced lambing in Awassi ewes.

## 1. Introduction

Small ruminants residing in temperate latitudes rely upon the photoperiod as a time signal to synchronize their reproductive activity with the external environment [1]. Changes in day length are perceived by the retina and translated into a chemical signal by the pineal gland through nocturnal secretion of melatonin [2]. Therefore, secretion and blood concentrations of melatonin are low during daylight hours and high at night-time; this is an organic medium that informs ruminants of photoperiods [3,4]. In mammalian reproduction, melatonin determines its effect at different levels of the hypothalamic, pituitary, and gonadal axis via specific receptors [5,6]. 

There are three melatonin receptor subtypes, which belong to the superfamily of G protein-coupled receptors [7]; these receptor subtypes are melatonin receptor 1A (MT1), melatonin receptor 1B (MT2), and melatonin receptor 1C (MT3). These subtypes’ coding genes have been cloned [8] and mapped in several animal species [9]. The melatonin receptor 1A (*MTNR1A*) gene is located on Chromosome 26 in sheep. It extends from the 17,354,820 to 17,377,973 position of the Oar_rambouillet_v1.0 genome assembly and consists of two exons, separated by an intron nearly 22 kilo-bases (kb) in length. The exons I and II of the *MTNR1A* gene are approximately 260 and 970 baise-pair (bp) in length, respectively. The *MTNR1A* locus exhibits several polymorphic sites, which are associated with seasonal reproductive activity in ewes [10,11], goats [12,13], buffalos [14], and wild animals [15]. Whilst the effects of the *MTNR1A* gene on reproductive seasonality studied so far are related to three specific mutations found in the exon II, at positions 606, 612, and 706 [16,17], the other parts of this gene have only been investigated in Sarda and Aragonese sheep [18]. These studies on the entire *MTNR1A* gene in these two breeds have provided confirmation of the influence of this gene in the control of reproductive seasonality and new knowledge useful for improving the reproductive management of sheep farms. Considering that the economy of many Mediterranean areas is greatly linked to the evolution of the sheep-farming system, it would be beneficial to characterize the *MTNR1A* gene sequence in the Awassi sheep, which is a very widespread breed in the Mediterranean area [19,20]. The opportunity to control and manage reproductive seasonality is a strong tool for economic growth in developing countries. Therefore, the first aim of this study was to identify the polymorphisms from the entire *MTNR1A* coding region and promoter in Lebanese Awassi sheep flocks. The second aim of this study was to investigate the association between the found polymorphisms and the reproductive performance of the sheep flocks.

## 2. Materials and Methods

### 2.1. Ethical Approval

Animals used in this research were managed and treated by the farm’s veterinary service in accordance with the European Commission’s recommendations. An explicit ethical agreement was not necessary as blood samples were taken by the National Health Veterinary Service during routine flock health control procedures. Sheep were included in this trial with the farmers’ permission.

### 2.2. Animals and Management

The study was conducted in two districts of Lebanon—Bekaa and Jezzine—located in the northeast and south of Lebanon respectively. In each district, a sheep farm was selected, each raising approximately 200 Awassi ewes that had been exposed to natural photoperiods since birth.

The Bekaa Plain stretches from Mount Lebanon to Anti-Lebanon, the altitude of which ranges from 600 to 1100 m. The plain is divided into two climatic regions: (1) the north with a sub-desert climate where the average annual temperature is 17 °C, with low rainfall ranging from 200 to 400 mm and (2) the south where the average annual temperature is 15 °C, with higher rainfall ranging from 900 to 1000 mm. Alternatively, in the Jezzine Coastal Plain the average annual temperature is 19 °C and the rainfall ranges from 700 to 1000 mm per year. The rainfall in Lebanon is notably high during the winter, and concentrated from November to February. In contrast, the summer is very dry, particularly in the north of the Bekaa. In both districts, the farming was of the semi-extensive type. Animals usually remain in and around the farm with low use of rangelands. Pastures are usually located near to the farm, and the flocks have short daily movement that allows them to return each evening to their farm. They depend on crop residue plots when available. In order to compensate for low forage quality and/or availability, and the negative effects induced by climatic conditions (summer drought from June to October and winter cold from December to January), ewes were provided with a daily cereal-based supplement. The most common food supplement for flocks in these farms included barley, yellow corn, and wheat, which were fed either whole or cracked. 

On 20 June 2017, 165 lactating ewes (89 in Bekaa district and 76 in Jezzine) aged 3–8 years (5.2 ± 1.5 years), with body condition score (BCS) 2.5–4.0 (3.3 ± 0.4), were chosen. Their BCS on a 1 to 5 scale [21] were assessed on June 10.

### 2.3. Reproductive Data Collection

From 1 July 2017 to 30 August 2017, the 89 ewes in Bekaa and the 76 ewes in Jezzine districts were, respectively, exposed to five and four adult rams. The health of the rams was confirmed by a veterinarian, and they were proven to be fertile as they had produced offspring in previous breeding seasons. The timing of the exposure ensured that lambing occurred between late November and the end of January. From 150 to 220 days after ram introduction, lambing dates and litter size were registered. From the recorded data, lambing rate (number of lambed ewes per ewe exposed to the ram), litter size (number of new-born lambs per lambed ewes), and days to lambing (DTL, i.e., number of days between the joining of rams until the subsequent lambing) were calculated.

### 2.4. Genotyping

From the jugular vein of each ewe, 10 mL of blood was collected using sterile vacuum tubes (BD Vacutainer System, Belliver Industrial Estate, Plymouth, UK) with ethylenediamine tetraacetic acid (EDTA) as an anticoagulant. 

DNA extraction protocols and all the used primers for the amplifications of the promoter genomic region, exon I, exon II, and partial 3′ UTR were those reported by Luridiana et al. [22]. The polymerase chain reaction (PCR) and the electrophoresis conditions were performed following Luridiana et al. [22]. All PCR products were sequenced in forward and reverse direction by a commercial service (BioFab Research, Roma, Italy). Information about primers used for fragment amplification and PCR conditions are available in Appendix A.

The found sequences were subjected to Nucleotide Blast (www.ncbi.nlm.nih.gov/blast/, accessed on 19 January 2021) for the alignment with the latest sheep genome assembly (Oar_rambouillet_v1.0—GenBank assembly accession number: GCF_002742125.1). 

Bioedit Sequence Alignment Editor software (http://www.mbio.ncsu.edu/BioEdit/BioEdit.html, 19 January 2021) was used to align nucleotides.

### 2.5. Statistical Analysis

Allele frequency was obtained by direct counting of the found genotypes. The chi-squared test was used to determine the Hardy–Weinberg equilibrium of the variations (Genepop 4.2). R statistical software (Version 4.0.0) [23] was used to analyze the associations between each genotype (three levels), and reproductive traits—namely, the lambing rate, litter size, and DTL. Haplotype association studies, by Rpackage Haplo.stats [24] were used to evaluate the reproductive performances in each of the identified haplotypes. The same linear model was used for genotypes and haplotypes:Y_jklm_ = µ + G_j_ + A_k_ + D_l_ + G_j_A_k_ + e_jklm_(1)
where Y_jklm_ is the trait measured for each animal, µ is the overall mean, G_j_ is the fixed effect of the genotype (j = 3 levels, the 2 homozygotes and the heterozygote) or haplotype (j = 5 levels, H1 to H5), A_k_ is the age of each animal (k = 6 levels, 3–8), D_l_ is the random effect of the farming district (l = 2 levels, Bekaa and Jezzine), G_j_A_k_ is the interaction between genotype or haplotype and age, and e_jkl_ is the random residual effect of each observation. To compare number of ewes lambed each 10 days, we used a χ^2^ test. Statistical significance was set at *p* < 0.05 using the Bonferroni method at α = 0.05.

## 3. Results

The obtained amplified and sequenced fragments corresponded to the exons I and II coding regions, to the promoter, to part of intron I, and the 3′ UTR region. From the sequence analysis, 31 single nucleotide polymorphisms (SNPs) were detected, and accordingly named SNP1–SNP31 (Table 1). The allele and genotype frequency, SNP identification code (ID), and *p*-values of the Hardy–Weinberg equilibrium are shown in Table 1. All genotype positions were inserted according to the last sheep genome assembly version (Oar_Rambouillet_v1.0, with accession RefSeq GCF_002742125.1). The identified SNPs were as follows: 15 in the promoter region; 1 in the intron I; 1 in the exon I; 10 in the exon II; and 4 in the 3′ UTR. Genotype distributions in SNP9 and SNP14 (promoter), in intron I, in exon I and II, and in 3′ UTR were in Hardy–Weinberg (HW) equilibrium. The other SNPs within the promoter region were not in equilibrium due to their high heterozygosity. On the other hand, SNP12 and SNP15 were in HW disequilibrium due to the low heterozygosity. Among all the SNPs, five were missense, which were within the exon II. 

All of the SNPs causing an amino acid change were predicted as tolerated by variant effect predictor (VEP); their sorting intolerant from tolerant (SIFT) scores (to predict the effect of an amino acid substitution on protein function) were as follows: SNP17 = 0.27; SNP21 = 0.37; SNP22 = 0.22; SNP24 = 0.12; and SNP25 = 1. The SNP20 (rs430181568) and SNP21 (rs407388227) were totally linked (D’ = 1 and r2 = 1), so that they can be considered as a single marker.

All the SNPs were analyzed to evaluate their relationship with lambing rate, litter size, and DTL. The overall mean and the standard deviation (SD) for DTL and litter size in the enrolled ewes were 168.6 ± 19.4 and 1.1 ± 0.15, respectively; the lambing rate of the studied population was 0.87. A graphical representation about the distribution of each phenotypic trait based on age of enrolled ewes is shown in the Appendix A. 

Statistical analysis showed that the C/C and T/C genotypes in SNP20 (rs430181568) and SNP21 (rs407388227) were associated with a shorter DTL, whilst litter size and lambing rate were not associated with the genotypes (Table 2). None of the other SNPs exhibited association with the analyzed reproductive traits. All the haplotypes with a frequency higher than 0.05 were considered in the haplotype association study for a total of five haplotypes, identified with the numerical code H1–H5, as shown in Table 3. The haplocodes H2, H3, and H4 were associated with lower DTL (*p* < 0.05) (Table 4). Litter size and lambing rate did not show statistical significance in the five haplotypes. 

## 4. Discussion

The Awassi sheep is the most widespread, medium-sized breed of the Middle East, raised for milk and meat production. Its importance derives mainly from the easy adaptation to the hard environmental conditions; they endure well in the high temperatures, dietary fluctuation, and parasites infestation. Under extensive conditions, the mating period occurs in early summer and first lambing after 20 months of age. The ewes enrolled in this study were 3 to 8 years old, so that they were at least at their second lambing.

The genetic variability emerging from the *MTNR1A* gene sequences in the Awassi ewes is near to that found in the Sarda breed, however, there are some differences from the Raza Aragonesa breed [18,22]. Thirty one SNPs were identified in this study, compared to the 29 SNPs identified in the Sarda breed thus far, and the differences were observed at exon II and the promoter level. The five polymorphic sites causing amino acid changes among Awassi ewes were the same as those reported among the Sarda breed [22]. Alternatively, Calvo et al. [18] detected 12 additional SNPs causing amino acid changes within the exon II among the Raza Aragonesa breed. The analysis of the Awassi exon II sequence showed that the SNP20 (old 612/MnlI; rs430181568) was always associated with the SNP21 (rs407388227), which was coherent with that found in other European breeds [17,25]. This suggests that one or both SNPs may be involved in the regulation of reproductive seasonality [26]. The SNP21 (rs407388227) was found to cause an amino acid change (Val > Ile) in position 220 of the protein chain (GenBank access number AAB17721.1). This change is in the fifth transmembrane domain (TM5), which is a crucial site for the MT1a receptor functionality [27]. Missense mutations within this domain lead to important changes in signal transmission [28,29]. Furthermore, significant differences in cAMP inhibition among sheep with Val220 and Ile220 were observed [16], thus suggesting a possible change in melatonin signal transmission in sheep carrying different alleles at SNP21 (rs407388227). These variations in the melatonin signal perception could lead to the phenotype differences in reproductive recovery observed in this study.

However, it should be considered that the sample size may have influenced the effect of SNPs on the analyzed phenotypic traits (DTL, litter size, and lambing rate). In particular, a larger sample size could evidence associations between other SNPs and the recorded reproductive traits, although in a previous study on another sheep breed using a higher number of ewes, no other associations were found [22]. Moreover, ewes carrying the T/T genotype at SNP20/SNP21 (rs430181568/rs407388227) delayed the onset of their reproductive activity compared to the C/C and C/T ewes. This delay led to the different DTL recorded in this study, suggesting that among the observed ewes, there were differences in sensitivity to photoperiod. Furthermore, the identification of melatonin receptors in granulosa and other ovary cells confirms that melatonin plays an important role in different ovarian functions [30]. In the preovulatory follicular fluid, the recorded melatonin concentrations were approximately threefold higher than in the bloodstream [31]. These melatonin levels are presumably blood-derived, although precursors and enzymes for the synthesis of melatonin are found also in the ovary, and therefore it cannot be excluded that this organ also cooperates in the production of indolamine [32,33]. Melatonin administration has been shown to significantly counteract the apoptosis of granulosa cells through its antioxidant activity [31,34]. Some of these effects of melatonin on the granulosa cells are carried out through the decrease in the expression of some pro-apoptosis genes, and with the increase in the expression of other anti-apoptosis genes [35]. Both MT1 and MT2 receptors have been detected in granulosa cells, suggesting that the above effects could be carried out through the binding of melatonin receptors [31]. Recent studies have shown that silencing of receptor MT1 increases follicular atresia by increasing the expression of the anti-apoptosis genes [36]. Furthermore, the same research reported that in granulosa cells the expression of the GPX4 and SOD1 genes was inhibited after the silencing of the MT1 receptor, thus indicating that this receptor mediates the positive effect of melatonin on the expression of antioxidant genes.

Progesterone is essential for a regular estrus cycle and for the maintenance of pregnancy after ovulation [37]. Furthermore, progesterone production is directly related to the size of the ovarian follicle, with the highest secretion of this hormone from the granulosa cells in the larger ovarian follicles [38]. Melatonin increases the diameter of the corpus luteum, therefore increasing plasma progesterone concentrations [39]. The presence of melatonin receptors in the ovarian follicle suggests that melatonin can exert a direct control on the ovary [7,40]. It has been reported that melatonin induces progesterone secretion in granulosa cells, which appears partly to be due to *MTNR1A*, as the silencing of this receptor significantly reduces progesterone production [36,41]. Therefore, the inhibitory effect of melatonin on apoptotic genes, and its stimulatory effect on antioxidant genes and progesterone production, carried out through the MT1 receptor, is able to influence reproductive activity in sheep. So, as the different polymorphisms in SNP20 (rs430181568) and SNP21 (rs407388227) can influence the melatonin signal transmission, as mentioned above, it follows that even at the ovarian level, this could change the reproductive response of ewes. The lower DTL shown by ewes carrying the C/C or C/T genotype at SNP21 could be carried out by melatonin through a potential antiapoptotic and antioxidant effect via the MT1 receptor. This effect could safeguard the follicle development, thus causing an advanced reproductive resumption. In contrast, the longer DTL recorded in T/T ewes might depend on a limited follicle development.

The data analysis additionally highlighted that the presence of at least one C allele was enough to determine a positive effect on reproductive recovery. These findings are coherent with previous research involving different breeds of sheep [42,43]. The haplotype analysis equally supports this finding, since H2, H3, and H4, which all exhibited the C allele at positions 20 and 21 (corresponding to SNP20-rs430181568 and SNP21-rs407388227, respectively), are associated with a shorter DTL.

The fertility rate was not affected by polymorphisms of the *MTNR1A* gene, although we expected at least SNP21 (rs407388227) would influence the fertility rate as registered in other breeds [17,26]. This fact leads to several considerations that could explain the lack of effect of this polymorphism on sheep fertility. The first consideration that must be made is that the effect of this polymorphism on reproductive activity is marginal in this breed of sheep. This fact agrees with what was reported by Hernandez et al. [44] in the Ile de France breed, where the SNP20 (rs430181568) (which is totally linked to the SNP21, rs407388227) did not influence the reproductive activity. These authors suggested that the influence of this polymorphism on the regulation of reproductive seasonality depends on the sheep breed and/or environmental conditions. Another hypothesis could be that in the present study the ram introduction in the flock occurred after the photoperiod inversion, so that the ewes could be in a shallower anestrous state and be more sensitive to the ram effect. Thus, in the present research, lambings began to occur around December 15 overall in the ewes carrying the C/C and C/T genotypes, indicating that these ewes were more ready to resume reproductive activity at the ram introduction. Over time, the effect of the photoperiod became stronger leading the T/T ewes to resume their reproductive activity. Therefore, at the end of the observations, all the genotypes showed the same fertility rate. 

## 5. Conclusions

This study provided the entire coding region of the *MTNR1A* receptor gene in the Awassi sheep breed. Thirty-one polymorphic sites were identified, five of which caused amino acid changes. The genotype and haplotype analysis highlighted that the presence of only one C allele in SNP20 (rs430181568) and SNP21 (rs407388227) is enough to advance reproductive recovery in this breed of sheep. Moreover, the above SNPs were totally linked and might be considered as markers for practical application in the selection of this trait. These results are therefore a valuable tool for improving reproductive management and obtaining advanced lambing in Awassi ewes. In order to better understand the effect of melatonin on the ovary, it would be useful to investigate whether the ewes carrying at least one C allele at SNP20 (rs430181568) and SNP 21 (rs407388227) show changes in the expression of genes promoting follicular development and the antioxidant effect at the ovarian level.

## Figures and Tables

**Table 1 animals-11-00583-t001:** Number and ID of single nucleotide polymorphisms (SNPs), genotype, allele frequency, Hardy–Weinberg equilibrium, locus, and amino acid (AA) change in the *MTNR1A* gene. The sequence is in a reverse orientation on the genome, and the SNPs are ordered according to their positions in the latest sheep genome version (Oar_Rambouillet_v1.0, GCF_002742125.1).

Alias	SNP ID	Position on Oar_rambouillet_v1.0	Genotype	Genotype Frequency	Allele	Allele Frequency	Hardy–Weinberg	*MTNR1A* Region	AA Change
SNP1	rs409468184	17379432	GG	62.9	G	0.815	0.023	Promoter	-
			GA	37.1	A	0.185			
			AA	0					
SNP2	rs428941001	17379252	CC	61.2	C	0.806	0.016	Promoter	-
			CA	38.8	A	0.194			
			AA	0					
SNP3	rs419424336	17379237	GG	66.6	G	0.833	0.045	Promoter	-
			GA	33.4	A	0.167			
			AA	0					
SNP4	rs429917252	17379178	CC	65.1	C	0.825	0.035	Promoter	-
			CT	34.9	T	0.175			
			TT	0					
SNP5	rs405080439	17379097	GG	61.9	G	0.809	0.019	Promoter	-
			GA	38.1	A	0.191			
			AA	0					
SNP6	rs428880789	17379083	GG	64.8	G	0.824	0.033	Promoter	-
			GA	35.2	A	0.176			
			AA	0					
SNP7	rs415456480	17379037	GG	62.1	G	0.811	0.019	Promoter	-
			GA	37.9	A	0.189			
			AA	0					
SNP8	rs406184829	17378992	CC	0	C	0.349	0.000	Promoter	-
			CT	69.8	T	0.651			
			TT	30.2					
SNP9	rs411931887	17378874	CC	63.0	C	0.778	0.152	Promoter	-
			CT	29.6	T	0.222			
			TT	7.4					
SNP10	rs402949406	17378871	CC	0	C	0.175	0.035	Promoter	-
			CA	34.9	A	0.825			
			AA	65.1					
SNP11	rs426266687	17378842	CC	66.8	C	0.834	0.047	Promoter	-
			CT	33.2	T	0.166			
			TT	0					
SNP12	rs399461430	17378769	CC	0	C	0.343	0.000	Promoter	-
			CT	68.5	T	0.657			
			TT	31.5					
SNP13	rs400561563	17378728	CC	55.5	C	0.703	0.004	Promoter	-
			CT	29.6	T	0.297			
			TT	14.9					
SNP14	rs419743392	17378706	GG	67.9	G	0.840	0.056	Promoter	-
			GA	32.1	A	0.160			
			AA	0					
SNP15	rs406334919	17378624	CC	0	G	0.176	0.033	Promoter	-
			CT	35.1	A	0.824			
			TT	64.9					
SNP16	ss213714057	17377903	GG	66.6	G	0.814	0.822	exon 1	-
			GA	29.6	A	0.186			
			AA	3.8					
SNP17	-	17377662	GG	3.2	G	0.187	0.854	intron 1	-
			GC	30.9	C	0.813			
			CC	65.9					
SNP18	rs419680097	17355611	CC	48.1	C	0.685	0.564	Exon 2	-
			CA	40.7	A	0.315			
			AA	11.2					
SNP19	rs406779174	17355458	GG	74.1	G	0.852	0.231	Exon 2	-
			GA	22.2	A	0.148			
			AA	3.7					
SNP20	rs430181568	17355452	CC	49.2	C	0.691	0.476	Exon 2	-
			CT	39.7	T	0.309			
			TT	11.1					
SNP21	rs407388227	17355358	CC	49.2	C	0.691	0.476	Exon 2	Ile/Val
			CT	39.7	T	0.309			
			TT	11.1					
SNP22	rs404378206	17355190	CC	92.5	C	0.963	0.697	Exon 2	Ile/Val
			CT	7.5	T	0.037			
			TT	0					
SNP23	rs429718221	17355173	GG	44.4	G	0.666	0.984	Exon 2	-
			GA	44.4	A	0.334			
			AA	11.2					
SNP24	rs403212791	17354971	GG	73.0	G	0.865	0.119	Exon 2	Cys/Arg
			GA	27.0	A	0.135			
			AA	0					
SNP25	rs426523476	17354963	GG	50.1	G	0.702	0.695	Exon 2	-
			GA	40.2	A	0.298			
			AA	9.7					
SNP26	rs413084140	17354943	CC	11.9	C	0.335	0.761	Exon 2	His/Arg
			CT	43.2	T	0.665			
			TT	44.9					
SNP27	rs403826495	17354935	CC	13.8	C	0.344	0.384	Exon 2	Ile/Val
			CT	41.2	T	0.656			
			TT	45.0					
SNP28	rs423194759	17354883	CC	41.3	C	0.661	0.272	3′Utr	-
			CT	49.7	T	0.339			
			TT	9.0					
SNP29	rs414185743	17354835	CC	14.2	C	0.370	0.827	3′Utr	-
			CT	45.6	T	0.630			
			TT	40.2					
SNP30	rs400830807	17354827	GG	74.2	G	0.871	0.139	3′Utr	-
			GA	25.8	A	0.129			
			AA	0					
SNP31	rs410686330	17354746	GG	85.3	C	0.927	0.428	3′Utr	-
			GA	14.7	T	0.073			
			AA	0					

**Table 2 animals-11-00583-t002:** Litter size, fertility rate, and days to lambing (DTL) for each genotype at SNP20/SNP21. Data are expressed as LSmeans.

Genotype	Litter Size	Lambing Rate (%)	DTL
C/C	1.2	88	164 ± 18.1 ^a^
C/T	1.1	87	170 ± 18.2 ^a^
T/T	1.1	87	184 ± 22.4 ^b^

Note: DTL = days to lambing; Different lower-case letters (a, b) indicate differences for *p* ≤ 0.05.

**Table 3 animals-11-00583-t003:** Haplotypes and their frequency in the enrolled ewes.

Haplocode	1	2	3	4	5	6	7	8	9	10	11	12	13	14	15	16	17	18	19	20	21	22	23	24	25	26	27	28	29	30	31	Haplotype Frequency
1	A	A	A	T	A	A	A	C	T	C	T	C	C	A	C	A	G	A	G	T	T	C	A	G	A	C	C	T	C	G	A	0.05470
2	A	A	A	T	A	A	A	C	T	C	T	C	C	A	C	A	G	C	A	C	C	C	G	A	G	T	T	C	C	A	G	0.05470
3	G	C	G	C	G	G	G	T	C	A	C	T	C	G	T	G	C	C	G	C	C	C	G	G	G	T	T	C	C	G	G	0.27695
4	G	C	G	C	G	G	G	T	C	A	C	T	C	G	T	G	C	C	G	C	C	C	G	G	G	T	T	C	T	G	G	0.14733
5	G	C	G	C	G	G	G	T	C	A	C	T	T	G	T	G	C	A	G	T	T	C	A	G	A	C	C	T	C	G	G	0.20156

**Table 4 animals-11-00583-t004:** Litter size, fertility rate, and DTL according to haplotypes. Data are expressed as LSmeans.

Haplocode	Litter Size	Fertility Rate (%)	DTL
1	1.1	82	183 ± 22.6 ^b^
2	1.0	84	166 ± 18.3 ^a^
3	1.1	87	161 ± 18.2 ^a^
4	1.0	88	167 ± 19.1 ^a^
5	1.0	81	182 ± 20.4 ^b^

Note: DTL = days to lambing; Different lower-case letters (a, b) indicate differences for *p* ≤ 0.05.

## Data Availability

The data presented in this study are available on request from the corresponding author. The data are not publicly available to preserve privacy of the data.

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
