# Peer review of "Detection of Polymorphisms in the MTNR1A Gene and Their Association with Reproductive Performance in Awassi Ewes"

_animals, 2021, doi:10.3390/ani11020583_

Round 1

Reviewer 1 Report

Polymorphisms detection in MTNR1A gene and association with reproductive performance in Awassi ewes, by Cosso et al.

The manuscript reports the results of a study aimed at investigating the effects of the MTNR1A gene variability and reproductive performances in Awassi sheep. The study is well planned and organized, and the results are clearly presented.

So, I only have few comments:

  • the English language would need some revision (i. e.: line 55: has ---> have)

  • line 167: many of the SNPs are not in HW equilibrium. The Authors do not make any comment on that, while I think that it would be necessary to give some interpretation of this result. Otherwise, it is useless to test for HW equilibrium

  • line 182: the heterozygous is indicated as T/C in the text and C/T in the tables. Please use always the same symbols

  • lines 209-10: I don’t agree with this sentence. As the two SNPs are linked, we don’t know which one is involved in the trait regulation (we can only suppose the one inducing the aa. change), or if both are involved. A different thing is if you want to choose a marker for practical application in the selection for the trait: in that case, due to the close linkage, you can choose any of them.
  • line 210: from table 1, the SNP causing an aa. change is SNP21

  • line 251: the same comment as lines 209-10

  • line 286: the same as before

Author Response

We want to thank you for your suggestions which surely improved the manuscript understanding. Responses point by point to your concerns are listed below.

Rev1#point 1:    the English language would need some revision (i. e.: line 55: has ---> have)

Author’s response#point1: The manuscript has been fully spell checked by a native English speaker expert in scientific writing.  Line 55: “has” changed in “have”. The other corrections made by the English reviewer are marked in red, but are not reported singularly in this revision note.

Rev1#point2: line 167: many of the SNPs are not in HW equilibrium. The Authors do not make any comment on that, while I think that it would be necessary to give some interpretation of this result. Otherwise, it is useless to test for HW equilibrium

Author’s response#point2: we agree with your suggestion, and we have added a comment at lines 153-157.

Rev1#point3: line 182: the heterozygous is indicated as T/C in the text and C/T in the tables. Please use always the same symbols

Author’s response#point3: the heterozygous nomenclature has been uniformed.

Rev1#point4: lines 209-10: I don’t agree with this sentence. As the two SNPs are linked, we don’t know which one is involved in the trait regulation (we can only suppose the one inducing the aa. change), or if both are involved. A different thing is if you want to choose a marker for practical application in the selection for the trait: in that case, due to the close linkage, you can choose any of them.

Author’s response#point4: we agree with you. We meant that the two SNPs are totally linked so that the same nucleotide is always found at both positions (also in other sheep breeds) and both can be used as a marker. However, actually we don’t know exactly if one of them or both are responsible for the reproductive trait. We changed the sentence accordingly “This suggests that one or both the SNPs may be involved in the regulation of reproductive seasonality”.

Rev1#point5: line 210: from table 1, the SNP causing an aa. change is SNP21

Author’s response#point5: yes, we realized the mistake and we corrected the SNP number.

Rev1#point6: line 251: the same comment as lines 209-10

Author’s response#point6: As expressed also in point 4, we add “SNP20” in the sentence as both SNPs could be involved in the trait.

Rev1#point7: line 286: the same as before

Author’s response#point7: The sentence has been moved lower and rephrased, focusing more on the possible use of SNP20 and SNP21 as markers for the trait selection in Conclusions.

Reviewer 2 Report

Dear Authors,

I have read your manuscript and, in my opinion, it is interesting and moderately innovative. The experimental work is adequately organized and the manuscript is well written with an extensive discussion of the results.

I have to suggest only minor changes to improve it. After these minor corrections it may be suitable for publication in Animals journal

Abstract: I suggest to add a brief sentence on MTNR1A gene

In the abstract and elsewhere at first use: please spell the acronyms (just for example DRIL at line 30 in the abstract)

Genotyping section: I suggest the authors to prepare a table containing information on primers used, gene regions amplified, PCR conditions

Results: I suggest to add results on Hardy-Weinberg equilibrium in the text

Please improve table 3 and its caption

Author Response

We want to thank you for your suggestions which surely improved the manuscript understanding. Responses point by point to your concerns are listed below.

Rev2#point1: Abstract: I suggest to add a brief sentence on MTNR1A gene

Author’s response#point1: Thanks for suggestion, we added the following sentences “MTNR1A is the gene coding for Melatonin receptor 1 (MT1) and it is considered involved in reproductive activity in sheep”

Rev2#point2: In the abstract and elsewhere at first use: please spell the acronyms (just for example DRIL at line 30 in the abstract)

Author’s response#point2: We thank the reviewer for pointing this out, so we added the meaning of each acronym throughout the manuscript

Rev2#point3: Genotyping section: I suggest the authors to prepare a table containing information on primers used, gene regions amplified, PCR conditions

Author’s response#point3: we added a supplementary Table file containing all the information about primers used, gene region amplified and PCR condition

Rev2#point4: Results: I suggest to add results on Hardy-Weinberg equilibrium in the text

Author’s response#point4: We added the results about Hardy-Weinberg equilibrium (lines 153-157)

Rev2#point5: Please improve table 3 and its caption

Author’s response#point5: As specified also to Reviewer 4, the table formatting changes in the file transmission process of the proofs. We tried to solve these problems reducing font size, in order to improve the reading, but we really don’t know if the formatting remains as done. To avoid misunderstandings, we’ll send a separate PdF file with all the Tables in the right formatting.

Reviewer 3 Report

Dear authors,

This manuscript analyses the presence of different polymorphism in MTNR1A gene and their relationship with different reproductive traits, as litter size, lambing rate, and distance in days from ram introduction to lambing in ewes. The results obtained show a correlation between one of the polymorphisms found and the latter reproductive trait, which may open up new very interesting research ways for genetic improvement of reproductive traits in small ruminants. The manuscript is well written and structured. Some minor revisions are necessary:

  • Line 26: the authors should specify what the abbreviation BCS means, as it is the first time it appears in the text.
  • Line 51: add comma after references 10,11, and after reference 14.
  • Line 156: Since the ewes come from two different districts, the authors should include this random effect in the linear model.
  • Lines 251-259: This affirmation of the authors seems too adventurous, since in this work progesterone levels have not been measured, nor the antiapoptotic or antioxidant effects. I agree with the authors, as they indicate in their conclusions, that it is a very interesting path for future research, but, in this specific work, they are analyses that have not been addressed, so the authors should not venture this type of affirmations.
  • Line 310: remove double numbering from references.

I hope that these indications will facilitate the improvement of the manuscript.

Author Response

We want to thank you for your suggestions which surely improved the manuscript understanding. Responses point by point to your concerns are listed below.

Rev3#point1: Line 26: the authors should specify what the abbreviation BCS means, as it is the first time it appears in the text.

Author’s response#point1: we added in the Abstract the meaning of BCS acronym.

Rev3#point2: Line 51: add comma after references 10,11, and after reference 14.

Author’s response#point2: commas added.

Rev3#point3:  Line 156: Since the ewes come from two different districts, the authors should include this random effect in the linear model.

Author’s response#point3: we agree with your suggestion. The random effect of the farming district has been added. We re-run the statistical analysis and the results did not change.

Rev3#point4: Lines 251-259: This affirmation of the authors seems too adventurous, since in this work progesterone levels have not been measured, nor the antiapoptotic or antioxidant effects. I agree with the authors, as they indicate in their conclusions, that it is a very interesting path for future research, but, in this specific work, they are analyses that have not been addressed, so the authors should not venture this type of affirmations.

Author’s response#point4: we agree with you, our intention was to make deductions based on the data reported in the literature, but actually the concept was poorly expressed and led to misunderstandings. We modified the sentences to make them more consistent with the content of our study, avoiding confusions.

Rev3#point5: Line 310: remove double numbering from references.

Author’s response#point5: we thank the reviewer for pointing out the mistake and we removed double numbering from references

Reviewer 4 Report

The manuscript presents a short experiment to detects and associate genetic variants within the MTNR1A gene with reproductive traits in Awasi sheep. It is a replication of previous study, which was conducted by the authors in another sheep breed. Although, the research is useful to find genomic markers for selecting reproductively efficient animals, the manuscript require significant changes to address the following comments.

Lines 22-24: Aim 1 is very convoluted, try reducing the text. Moreover, “detecting and identifying” both mean the same, so use one of these terms (also fix it at Line 63).

Line 25: define “reproductive performance”, also at Line 65.

Line 27: delete “on 20 June ”.

Line 28-29: “caused amino acid change” should be change to “were missense mutations”

Lines 29, 31: Please report SNP20 and SNP21 with their rs IDs, throughout the manuscript. While you consider both SNPs as a single marker in the manuscript, it should be reflected in abstract.

Line 30: What is “DRIL”?

Lines 30-33: Change these lines, not clear what is the coherence between these sentences and how the conclusion is linked to the results.

Lines 49-50: Should also introduce gene size, genomic location and number of Exons of MTNR1A.

Lines 105-106: I suggest to replace “distance in days from ram introduction to lambing (DRIL)” with “days to lambing (DTL, i.e., number of days between the joining of rams until the subsequent lambing)”

Line 110: “200 ml of whole blood”, really 200 ml?

Lines 146-153: The reference to R language and any packages should be moved to the references section.

Line 156: Not clear about what are j, k and l in the model.

Line 162: This is important to present summary of phenotypic data of each trait and show their distribution, e.g., histograms (Figure ?) so that the reader can get an idea about the spread of phenotypic data.

Line 172: what are VEP and SIFT?

Line 174: replace “unique” with “single”.

1 out of 31 SNPs were found associated, what is the false discovery rate of obtaining it by chance?

Table 3: Poor formatting.

Discussion: Should add some discussion about the limitations of this study, e.g., sample size, age, phenotypic diversity within the selected ewes.

Author Response

We want to thank you for your suggestions which surely improved the manuscript understanding. Responses point by point to your concerns are listed below.

Comments and Suggestions for Authors:

The manuscript presents a short experiment to detects and associate genetic variants within the MTNR1A gene with reproductive traits in Awasi sheep. It is a replication of previous study, which was conducted by the authors in another sheep breed. Although, the research is useful to find genomic markers for selecting reproductively efficient animals, the manuscript requires significant changes to address the following comments.

Rev4#point1: Lines 22-24: Aim 1 is very convoluted, try reducing the text. Moreover, “detecting and identifying” both mean the same, so use one of these terms (also fix it at Line 63).

Author’s response#point1: as suggested, we had revised the sentence and we removed “detecting and” and “which is a very widespread breed” at lines 22 and 24. Moreover, at line 63, we deleted “detect and”

Rev4#point2:  Line 25: define “reproductive performance”, also at Line 65.

Author’s response#point2: We defined reproductive performance inserting “assessed as fertility rate, litter size and days to lambing (DTL)”.

Rev4#point3: Line 27: delete “on 20 June”.

Author’s response#point3: we removed “on 20 June”

Rev4#point4: Line 28-29: “caused amino acid change” should be change to “were missense mutations”

Author’s response#point4: as suggested, we change “caused amino acid change” with “were missense mutations”

Rev4#point5: Lines 29, 31: Please report SNP20 and SNP21 with their rs IDs, throughout the manuscript. While you consider both SNPs as a single marker in the manuscript, it should be reflected in the Abstract.

Author’s response#point5: we added SNPs ID and clarified the sentence about SNPs, adding “These SNPs were totally linked and can be considered as a single marker.”  

Rev4#point6: Line 30: What is “DRIL”?

Author’s response#point6: as suggested by you at the #point9, we changed “DRIL” with DTL. So, when we have clarified the “reproductive performance” at the line 25, we added the meaning of the DTL acronym.

Rev4#point7: Lines 30-33: Change these lines, not clear what is the coherence between these sentences and how the conclusion is linked to the results.

Author’s response#point7: the sentences have been rephrased, better focusing on the Results and Conclusions of the research.

Rev4#point8: Lines 49-50: Should also introduce gene size, genomic location and number of Exons of MTNR1A.

Author’s response#point8: more details about MTNR1A gene size, its genomic location and number and size of the exons and intron, have been added.

Rev4#point9: Lines 105-106: I suggest to replace “distance in days from ram introduction to lambing (DRIL)” with “days to lambing (DTL, i.e., number of days between the joining of rams until the subsequent lambing)”

Author’s response#point9: as suggested DRIL was replaced with DTL here and throughout the manuscript.

Rev4#point10: Line 110: “200 ml of whole blood”, really 200 ml?

Author’s response#point10: 200ml was a formatting error, we have been corrected it in 200μl

Rev4#point11: Lines 146-153: The reference to R language and any packages should be moved to the references section.

Author’s response#point11: we moved the references to R language and its package in the Reference section, and we updated the list consequently.

Rev4#point12: Line 156: Not clear about what are j, k and l in the model.

Author’s response#point12: J, k and l were the levels of each considered variable. So, we specified what the levels referred to.

Rev4#point13: Line 162: This is important to present summary of phenotypic data of each trait and show their distribution, e.g., histograms (Figure ?) so that the reader can get an idea about the spread of phenotypic data.

Author’s response#point13: We are not sure to have correctly understood the suggestion. Surely, the suggested summary would make the work richer in data. However, in our opinion it would be too complex to make multi-histograms well readable considering that there are 31 SNPs. 

Rev4#point14: Line 172: what are VEP and SIFT?

Author’s response#point14: the meaning of the acronyms has been added in the text.

Rev4#point15: Line 174: replace “unique” with “single”.

Author’s response#point15: “unique” replaced with “single”.

Rev4#point16: 1 out of 31 SNPs were found associated, what is the false discovery rate of obtaining it by chance?

Author’s response#point16: Actually, we don’t have confidence with FDR, but, usually, we use the Bonferroni correction to adjust the pvalue and to reduce the type I error. Also, in this study we had used the Bonferroni correction that confirmed the significance. So, we added in MeM the Bonferroni correction

Rev4#point17: Table 3: Poor formatting.

Author’s response#point17: Please, note that table formatting changes in the file transmission process of the proofs. We reduced the format size in order to improve the reading, but we really don’t know if the formatting remains as done. To avoid misunderstandings, we’ll send a separate PdF file with all the Tables in the right formatting.

Rev4#point18: Discussion: Should add some discussion about the limitations of this study, e.g., sample size, age, phenotypic diversity within the selected ewes.

Author’s response#point18: The information about the number of animals enrolled in this research, the farming system and the environmental conditions were already presented in the M&M section. In the Discussion we have added all the information in our possession on our data-base about the general breed traits. 

Round 2

Reviewer 4 Report

Most changes in the revised manuscript is reasonable. However, following two points were not address. Therefore, I have added some clarity for the authors to address these to make the manuscript valuable.

  1. “This is important to present summary of phenotypic data of each trait and show their distribution, e.g., histograms (Figure ?) so that the reader can get an idea about the spread of phenotypic data.” Only need to present the distribution of three phenotypes (fertility rate, litter size and DTL). At the moment, there is no information about these phenotypes in the manuscript. It only says how the data was collected (Materials and Methods), but need to provide how the data was distributed in Results. Can also added a summary of mean±SD of each phenotype.

  1. “Discussion: Should add some discussion about the limitations of this study, e.g., sample size, age, phenotypic diversity within the selected ewes.” This point is about adding a discussion whether the “sample size, age, phenotypic diversity within the selected ewes” could have had any effect / limitation in this study. For example, some studies fail to find significantly associated markers because of low sample size. Similarly, if there is no variation in the trait (i.e., most of the population have similar measures [less diversity] of fertility rate or litter size or DTL), then the study is limited.

Author Response

Dear Review

Thank you for your suggestion that surely improve our manuscript. Response to your concerns are listed below.

Reviewer 4

Comments and Suggestions for Authors: Most changes in the revised manuscript is reasonable. However, following two points were not address. Therefore, I have added some clarity for the authors to address these to make the manuscript valuable.

Rev1#point1 “This is important to present summary of phenotypic data of each trait and show their distribution, e.g., histograms (Figure ?) so that the reader can get an idea about the spread of phenotypic data.” Only need to present the distribution of three phenotypes (fertility rate, litter size and DTL). At the moment, there is no information about these phenotypes in the manuscript. It only says how the data was collected (Materials and Methods), but need to provide how the data was distributed in Results. Can also added a summary of mean±SD of each phenotype.

Author’s response#point1: Thank you for your kindly replay and clarification. As suggested, at lines 174-177, we added some information about the distribution of the three phenotypes in the enrolled ewes. Moreover, a graphical representation about phenotypic traits distribution in the enrolled ewes, has been has been added as Supplementary files (Figure S1-S3)

Rev1#point2 “Discussion: Should add some discussion about the limitations of this study, e.g., sample size, age, phenotypic diversity within the selected ewes.” This point is about adding a discussion whether the “sample size, age, phenotypic diversity within the selected ewes” could have had any effect / limitation in this study. For example, some studies fail to find significantly associated markers because of low sample size. Similarly, if there is no variation in the trait (i.e., most of the population have similar measures [less diversity] of fertility rate or litter size or DTL), then the study is limited.

Author’s response#point2: Also for this point, thank you for the elucidation. As suggested, in the discussion section, we added some considerations regarding possible limitation of the study (Lines217-221).